# RECALL-FIRST MODERATION VIA DISTRIBUTION-PRESERVING AUGMENTATION AND COMMITTEE-DIVERSE RETRIEVAL

## ABSTRACT

False negatives—missed unsafe content—remain the dominant risk in safety-critical moderation. We present a novel recall-first moderation framework that integrates two complementary innovations: (i) distribution-preserving contrastive augmentation, which generates boundary-focused hard positives and negatives while statistically preserving corpus structure, and (ii) committee-diverse retrieval, which combines dense, MMR, and graph-based selectors to construct label-informative, non-redundant neighborhoods at inference. Augmented corpora are validated with KL/JS divergence thresholds ($\leq 0.05$ globally), confirming indistinguishability from the source distribution. On a large held-out test set of multidomain unbalanced text, vanilla retrieval-augmented pipelines expose the persistent failure mode of under-detecting `FLAGGED` content (recall $\approx 0.44$), but also reveal a strong baseline gap: an open-source stack (FAISS + local LLaMA-3) achieves significantly higher accuracy and macro-F1 than a commercial counterpart (API embeddings + hosted LLM). Adding augmentation and committee retrieval improves sensitive-class recall by $\sim 10$ points (to $\approx 0.56$) while maintaining global performance, with graph-aware retrieval pushing open-source accuracy to 0.8510 and Macro-F1 to 0.7635. Ensemble experiments with DistilRoBERTa further raise recall to 0.5781 without loss of utility.

## 1 INTRODUCTION

Automated content moderation has become a critical application area for large language models (LLMs), due to their ability to interpret nuanced context and language. Recent studies have explored using LLMs to assist or even replace traditional classifiers in detecting harmful or policy-violating content (Huang, 2025; Chen et al., 2024a). Compared to static fine-tuned models, LLMs offer greater flexibility and understanding in borderline cases. OpenAI, for example, has proposed using GPT-4 as a content moderator to achieve more consistent policy enforcement (OpenAI, 2023). However, deploying LLMs for moderation also introduces new challenges: LLMs may reflect majority cultural biases (Nguyen et al., 2025), struggle with ambiguous "hard cases" that require contextual judgment (Huang, 2024), and can be resource-intensive to query repeatedly (Ding et al., 2024). Critically, a key risk in *safety-critical moderation* is *false negatives*—failing to flag harmful content. Recent work emphasizes the need to prioritize recall for unsafe content, even if it means tolerating some false positives (Chen et al., 2024a; Huang, 2024). Our work addresses this need by combining *retrieval augmentation* and *contrastive data augmentation* to boost the detection of subtle policy violations while maintaining overall accuracy in imbalanced datasets, as real-world data is typically distributed this way (He & Garcia, 2009; Chen et al., 2024b).

**Main Contributions.** We propose a novel LLM-based moderation pipeline that integrates *diverse retrieval* and *contrastive augmentation* to enhance recall of harmful content. First, we introduce a distribution-preserving paraphrase augmentation strategy that generates additional training examples which are statistically indistinguishable from the original data distribution (Papakipos & Bitton, 2022). Unlike standard augmentation (e.g., synonym replacement or back-translation (Feng et al., 2021)), our method produces paraphrases that closely match the original length and semantics but include *hard positives/negatives*—subtle rephrases of flagged content and safe content—to better expose decision boundaries. This approach builds on the idea of counterfactual and adversarial

augmentation in NLP (Kaushik et al., 2020; Hartvigsen et al., 2022), but focuses on *safety-critical* categories. Second, we design a *committee-based retrieval* mechanism to supply the LLM with diverse, relevant context at inference time. Instead of retrieving nearest neighbors with a single embedding model, our pipeline combines multiple retrievers and reranking strategies—dense semantic search, Maximal Marginal Relevance (MMR) diversification, and graph-based selection—to ensure retrieved examples are informative and non-redundant (Zhang et al., 2025; Chernogorskii et al., 2025; Rezaei & Dieng, 2025). Finally, we implement a *recall-oriented decision policy*: the model casts votes across multiple retrieved contexts, and we apply a calibrated threshold that leans on the side of flagging whenever there is reasonable doubt. By aggregating judgments and lowering the threshold for the positive class, we explicitly favor high recall for unsafe content—a property critical for safety filters (Huang, 2024). To our knowledge, this is the first framework to unify *distribution-preserving augmentation* with *retrieval-diverse RAG* for moderation, showing that careful retrieval diversity and thresholding can push an LLM moderator into a safer operating regime. In addition to its methodological contributions, we use the framework to *benchmark commercial RAG systems against open-source implementations* and to conduct a controlled comparison of open-source versus proprietary subscription-based models under a unified protocol.

## 2 RELATED WORK

Our work integrates three complementary research directions—LLM-based moderation, retrieval augmentation, and contrastive data augmentation—into a unified framework that explicitly prioritizes recall in safety-critical moderation scenarios. In this section we examine how prior work has applied these approaches in practice.

**LLMs for Content Moderation.** Recent studies increasingly explore the use of large language models (LLMs) in content moderation (Huang, 2024; OpenAI, 2023). Chen et al. (2024a) introduce CLASS-RAG, which enhances robustness against adversarial prompts by retrieving safe and unsafe examples for classification. Kolla et al. (2024) demonstrate how GPT-based assistants can support human moderators, while Franco et al. (2025) investigate workflows where LLMs generate policy-grounded explanations. At the same time, concerns remain regarding cultural bias and the dominance of majority perspectives in LLM moderation (Nguyen et al., 2025). To address this, Nguyen et al. (2025) propose *Mod-Guide*, a retrieval-augmented system designed to surface minority viewpoints. Our approach complements these efforts by placing explicit emphasis on recall-oriented moderation.

**Retrieval-Augmented Classification and Diversity.** Retrieval-augmented generation (RAG) has become central to knowledge-intensive NLP tasks (Izacard & Grave, 2020; Lewis et al., 2020). In moderation, retrieval provides LLMs with contextual policy text or representative examples (Chen et al., 2024a). However, the effectiveness of retrieval hinges on both relevance and diversity. Techniques such as maximal marginal relevance (MMR) (Carbonell et al., 1998), clustering-based selection (Zhang et al., 2025), and ensemble retrievers (Chernogorskii et al., 2025; Rezaei & Dieng, 2025) have been shown to improve coverage. Zhang et al. (2025) demonstrate that redundancy undermines recall, while Chernogorskii et al. (2025) introduce *DRAGON*, a retrieval training method that explicitly promotes diversity. Building on these insights, we employ an inference-time ensemble that integrates dense, MMR, and graph-based retrieval to ensure coverage of multiple facets of potentially harmful content.

**Data Augmentation for Moderation.** Data augmentation has proven effective for improving generalization, particularly under conditions of class imbalance. Traditional methods include synonym replacement, back-translation, and noise injection (Feng et al., 2021). More recently, generative augmentation with LLMs has been investigated (Ding et al., 2024; Kolla et al., 2024), though unconstrained paraphrasing risks shifting away from the original distribution. Papakipos & Bitton (2022) emphasize the importance of maintaining realism in augmented data. Alternative strategies, such as counterfactual augmentation (Kaushik et al., 2020) and adversarial generation for toxicity detection (Hartvigsen et al., 2022), have demonstrated strong improvements in recall. Extending this line of work, we introduce a lightweight paraphrase generator that produces *hard positives and negatives*, thereby enhancing boundary-focused exposure for classifiers.

# 3 METHODS

## 3.1 OVERVIEW

We propose a recall-oriented moderation framework that integrates three complementary components into a unified retrieval-augmented classification (RAG) pipeline. First, *distribution-preserving contrastive augmentation* generates boundary-focused examples that remain statistically indistinguishable from the original data, thereby strengthening decision boundary exposure. Second, *committee-based retrieval* leverages diverse retrieval strategies (dense, MMR, and graph-based) to construct more informative and less redundant neighborhoods at inference time. Finally, a *two-class ensemble layer based on DistilRoBERTa* is trained with classifier blending and a rank+veto fusion strategy, providing robust downstream classification while preserving high recall on safety-critical cases. Figure 1 provides a high-level illustration of the framework architecture and its core components.

**Recall-Oriented Moderation Framework**

Figure 1: **Overview of our recall-oriented moderation framework.** The system integrates *distribution-preserving contrastive augmentation* and *committee-based retrieval* into a unified RAG pipeline. On the left, augmented samples are generated and validated to preserve the original distribution. On the right, a committee of diverse retrievers ensures retrieval coverage across multiple facets of potentially harmful content.

## 3.2 DATA AUGMENTATION & DISTRIBUTION VALIDATION

We aim to generate label-preserving paraphrases that remain *statistically indistinguishable* from the original texts. To achieve this, indistinguishability is enforced at three complementary levels: global surface form, measured via word and character histograms; global semantic mixture proportions, captured in a compact embedding space; and local per-cluster consistency, where form is conditioned on latent semantics. Full algorithmic details are given in Appendix Section A.2. The overall augmentation and validation workflow is illustrated in Figure 2, which shows how embedding, clustering, constrained paraphrasing, and multi-level validation are combined to ensure distribution-preserving generation.

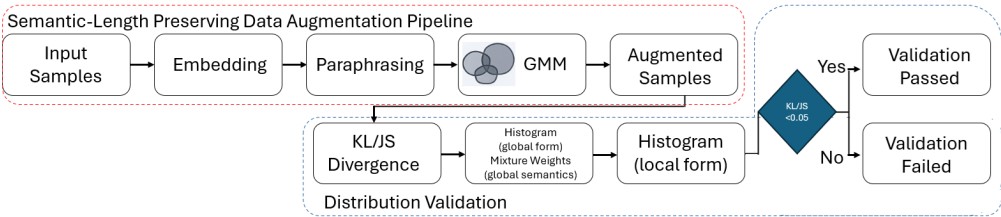

Figure 2: **Overview of the data augmentation and validation workflow**: paraphrased samples are generated under semantic- and structure-preserving constraints and validated through multi-level KL/JS divergence checks to ensure distributional fidelity.

The augmentation process begins by embedding texts, reducing dimensionality, and clustering them with a Gaussian Mixture Model (GMM). Augmented samples are then produced to satisfy integer quotas defined over the joint *class×component* distribution, ensuring that both label priors and latent structure are preserved. Rounding heuristics are applied to guarantee quota completion, with precise formulas provided in Appendix Section A.2.

Candidate paraphrases for each seed text are constrained to prevent semantic drift. A candidate $s'$ is only accepted if it satisfies both length and similarity requirements:

$$\text{len\_ok}(s, s') \ \wedge \ \text{sim}(s, s') \in [\ell, h],$$

where relative word and character deviations are bounded by $(\tau_w, \tau_c)$ and cosine similarity lies within $[\ell, h]$. Default windows and tolerance caps are reported in Appendix Section A.4. Additional mechanisms, including acceptance–relaxation rules, per-seed production and duplication limits, and starvation-prevention guard rails, further regulate the process to ensure robust coverage.

To validate distributional fidelity, we compute Kullback-Leibler (KL) and Jensen–Shannon (JS) divergence across three views: pooled-edge histograms of word and character counts (global form), mixture weights from a GMM fit on the originals only (global semantics), and per-component histograms (local form). Any global or per-component JS divergence exceeding $0.05$ is flagged as violating our indistinguishability tolerance. Implementation parameters include 30 bins, $99.5\%$ clipping, a reduced dimensionality of 50 via SVD, and $k = 5$ mixture components, as detailed in Appendix Section A.3.

Augmentation is deemed *distribution-preserving* only when all JS metrics—global form, global mixture proportions, and per-component maxima—are $\leq 0.05$. Mini-batch vectorization and cosine scoring yield costs that scale linearly with batch size, and runs terminate once all quotas are satisfied. Each accepted paraphrase preserves its lineage through a pointer and receives a stable identifier. To ensure reproducibility, we fix seeds, expose all hyperparameters via a command-line interface, and log both JSON and Markdown validation reports, with artifacts and exact commands documented in Appendix Section A.11.

### 3.3 RETRIEVAL-AUGMENTED PIPELINES

We design retrieval-augmented classification pipelines with strict controls to avoid data leakage and to enable fair comparison between commercial and open-source systems. The process begins by constructing train and test partitions exclusively from the original data. Augmented rows are then assigned to the split of their corresponding parent instance, ensuring that paraphrases never cross the train–test boundary. We summarize the overall architecture of these pipelines in Figure 3, which contrasts the commercial and open-source implementations under a unified evaluation framework.

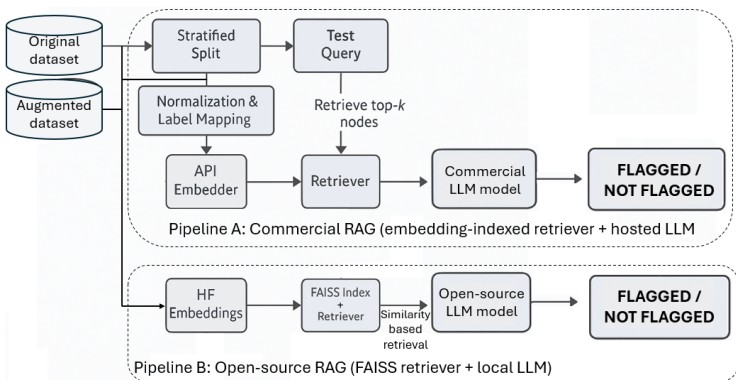

Figure 3: **Overview of the retrieval-augmented classification pipelines**: The pipeline prepares non-leaking data for two alternatives and evaluates them under a unified protocol.

Formally, stratified splits are created only from the original data, with each augmented sample $x_j^{\text{aug}}$ inheriting the partition of its parent $\pi(x_j^{\text{aug}})$, thereby enforcing a strict leakage-safe protocol:

$$\mathcal{D}_{\text{orig}}^{\text{train}}, \mathcal{D}_{\text{orig}}^{\text{test}} = \texttt{StratifiedSplit}(\mathcal{D}_{\text{orig}}, \texttt{test\_size} = 0.1), \quad x_j^{\text{aug}} \mapsto \text{split of } \pi(x_j^{\text{aug}}).$$

normalization and label mapping into $\{$FLAGGED, NOT FLAGGED$\}$ follow the preprocessing steps in Appendix Section A.1. The final training and test sets are therefore given by $\mathcal{D}^{\text{train}} = \mathcal{D}^{\text{train}}_{\text{orig}} \cup \mathcal{D}^{\text{train}}_{\text{aug}}$ and $\mathcal{D}^{\text{test}} = \mathcal{D}^{\text{test}}_{\text{orig}} \cup \mathcal{D}^{\text{test}}_{\text{aug}}$.

Once splits are fixed, both pipelines are evaluated under a unified retrieval-augmented classification protocol. For each test query $q$, a retriever $R_k(q)$ returns the top-$k$ nodes $\{n_1, \ldots, n_k\}$, which are injected alongside the query into a policy-first prompt. The model must then output a binary decision $\hat{y}(q) \in \{$FLAGGED, NOT FLAGGED$\}$. Retrieval quality is quantified using Hit@k, Precision@k, and normalized Discounted Cumulative Gain (nDCG@k), where

$$\text{DCG@}k = \sum_{i=1}^{k} \frac{2^{g_i} - 1}{\log_2(i+1)}, \quad \text{nDCG@}k = \frac{\text{DCG@}k}{\text{IDCG@}k},$$

and retrieval diversity is measured as $1 - \overline{\cos}$ (further details appear in Appendix Section A.5).

Pipeline A represents a commercial setup that relies on API-provided embeddings and hosted inference. Sentence-level chunks and the full policy are indexed using a 1536-dimensional API embedder, with retrieval based on similarity search . The classification step is carried out by a hosted instruction-tuned model (gpt-4o-mini, temperature 0.0), which is constrained to emit a single token in $\{$FLAGGED, NOT FLAGGED$\}$. Indices pin the embedder configuration to ensure dimension consistency, and implementation-specific parameters are documented in Appendix Section A.5.

Pipeline B, in contrast, is an open-source variant that leverages local infrastructure. A FAISS index is constructed over Hugging Face sentence embeddings (all-MiniLM-L6-v2), with policy text injected solely through the prompt. Inference is handled by a quantized local model, Power-LLaMA-3-7B-Instruct, executed via llama.cpp with a context window of approximately 4096 tokens and temperature fixed at 0.0. Retrieval uses a default of $k$=15. Engineering choices for this setup are also detailed in Appendix Section A.5.

The two pipelines therefore differ in critical ways. The commercial system integrates policy nodes directly into the index, employs API embeddings, and relies on hosted inference. The open-source system indexes only examples, uses Hugging Face embeddings, and performs inference locally.

Evaluation emphasizes recall on sensitive content. The primary metric is $\text{Recall}_{\text{FLAGGED}} = \frac{\text{TP}}{\text{TP+FN}}$, with secondary metrics including accuracy, macro- and weighted-F1, and macro-F2, along with the retrieval diagnostics described above. Statistical uncertainty is quantified via bootstrap confidence intervals, while significance testing employs McNemar's test, as detailed in Appendix Section A.7.

### 3.4 IMPROVING SAFETY-CRITICAL RECALL WITH CONTRASTIVE AUGMENTATION AND COMMITTEE-BASED RETRIEVAL

The dominant failure mode in safety-critical moderation remains false negatives, where harmful content escapes detection. To address this challenge, we introduce two complementary components. The first is **contrastive augmentation**, a mechanism designed to expose the decision boundary by generating boundary-focused hard positives and hard negatives while avoiding semantic drift. The second is a **committee-based retrieval** strategy, which replaces reliance on a single neighborhood of evidence with a diverse and balanced set of neighbors, thereby increasing the probability of label-diagnostic retrievals.To visualize how these components are orchestrated within our system, Figure 4 presents a detailed overview of the architecture, showing how contrastive augmentation and committee-based retrieval interact within the recall-first classification pipeline.

We formalize these intuitions through three hypotheses. **H1 (Boundary Exposure)** predicts that even small augmentation sizes, denoted $|\tilde{\mathcal{A}}|$, should yield measurable gains in recall, such that $\partial \text{Recall}_{\text{FLAGGED}}/\partial|\tilde{\mathcal{A}}| > 0$. **H2 (Evidence Diversity)** asserts that retrieval diversity increases the likelihood of observing at least one informative neighbor, quantified as $\Pr[\exists r : \text{Info}(N_r(x)) \geq \tau_{\text{diag}}] \uparrow$. Finally, **H3 (Thresholding)** hypothesizes that setting a recall-oriented voting threshold $\tau < 0.5$ on the committee's aggregate prediction $\hat{p} = \frac{1}{C}\sum_{c=1}^{C} \mathbf{1}\{\hat{y}^{(c)} = \text{FLAGGED}\}$ can produce Pareto improvements in recall with only limited accuracy trade-offs.

The contrastive augmentation module operationalizes H1. For each FLAGGED seed, we generate $p$ hard positives that remain within the FLAGGED class and $p$ hard negatives that are explicitly safe yet

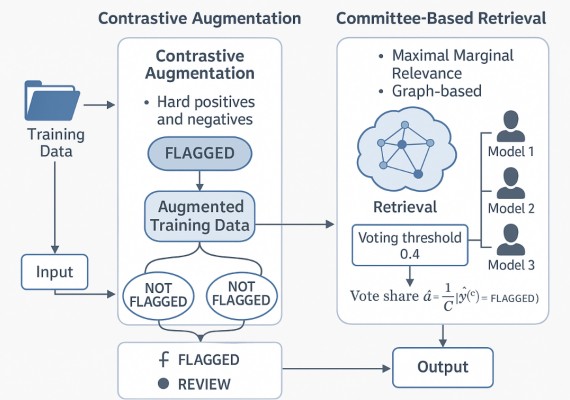

Figure 4: **Overview of Recall-Oriented Classification Framework Enhanced by Contrastive Augmentation and Committee-Based Retrieval:** The figure illustrates our dual-module enhancement: (i) *contrastive augmentation* generates semantically faithful hard positives and negatives to better expose decision boundaries, and (ii) *committee-based retrieval* leverages diverse retrievers to improve the chance of retrieving label-informative neighbors.

semantically close to the boundary. Each augmented row is linked to its parent instance and inherits the parent's train–test assignment to preclude leakage. Deterministic fallbacks and de-duplication strategies are applied to ensure both coverage and semantic stability.

Building on H2, committee-based retrieval assembles a pool of complementary retrievers. Specifically, we combine dense nearest-neighbor retrieval, Maximal Marginal Relevance (MMR) reranking, and a graph-aware selector that optimizes a multi-objective score. This score balances similarity to the query, penalization of redundancy, and a label-balance bonus, formally expressed as

$$\text{score}(x; q, S) = \alpha \sin(q, x) - \beta \operatorname{redundancy}(x, S) + \gamma \operatorname{balance\_bonus}(\ell_x).$$

Candidate pool construction and default settings are described in Appendix Section A.5. To convert committee outputs into predictions, we compute the vote share $\hat{p} = \frac{1}{C} \sum \mathbf{1}[\texttt{FLAGGED}]$ and predict FLAGGED whenever $\hat{p} \geq \tau$. For a recall-first orientation, we use $\tau = 0.40$.

Finally, both Pipeline A and Pipeline B integrate these components within the broader classification framework. Pipeline A uses API-provided embeddings, a chunked index, a hosted large language model, and retrieval with $k = 10$. Pipeline B employs Hugging Face embeddings, a FAISS index, a quantized local LLaMA model, and retrieval with $k = 15$. Despite these infrastructural differences, both pipelines share the same augmented training data, committee-based retrieval, recall-oriented vote thresholding, and evaluation protocol. Hyperparameters and logging conventions are consolidated in Appendix Section A.4, while reproducible artifacts are archived in Appendix Section A.11.

### 3.5 TWO-CLASS ENSEMBLE PIPELINE WITH DISTILROBERTA TRAINING, CLASSIFIER BLENDING, AND RANK+VETO GRID SEARCH

Our ensemble pipeline begins with the fine-tuning of `distilroberta-base` on JSONL data where labels are defined over two classes, $y \in \{0, 1\}$, with FLAGGED mapped to 1. Training optimizes cross-entropy loss, optionally with class-weighting, and the best checkpoint is selected using macro-F1. At inference time, the classifier outputs the probability $p_1 = p(y{=}1 \mid x)$. To enhance robustness, we introduce a blending strategy in which the predictions of two classifiers are averaged at the probability level. In the default configuration, the two classifiers are identical, so blending reduces to a no-op, though the framework allows for heterogeneous models. Full details of training and inference procedures are provided in Appendix Section A.8. We illustrate the overall architecture of this ensemble design in Figure 5, highlighting how blending and Rank+Veto fusion are integrated into the pipeline.

To further integrate external systems, we introduce a Rank+Veto ensemble mechanism that fuses the blended classifier with an external system $\mathcal{B}^+$. Fusion is based on a rank-dominant scoring rule augmented with a confidence veto. Specifically, for each instance $i$, the combined score is computed

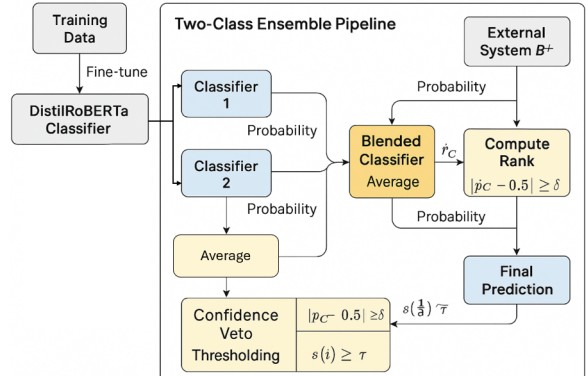

Figure 5: **Overview of the two-class ensemble pipeline:** The architecture combines fine-tuned DistilRoBERTa classifiers with probability-level blending and a Rank+Veto fusion strategy.

as

$$s(i) = 0.85\big[(1-\alpha)\tilde{r}_B(i) + \alpha\tilde{r}_C(i)\big] + 0.15\big[(1-\alpha)p_B(i) + \alpha p_C(i)\big],$$

where $\tilde{r}_B(i)$ and $\tilde{r}_C(i)$ denote normalized ranks, and $p_B(i)$ and $p_C(i)$ are the associated probabilities. Decision control is delegated to the classifier $\mathcal{C}$ when its confidence margin satisfies $|p_C - 0.5| \geq \delta$, while otherwise predictions are obtained by thresholding $s(i)$ at $\tau$. The default parameterization is $(\alpha, \tau, \delta) = (0.8, 0.6, 0.18)$, but these values are systematically tuned using grid search. The search ranges, as well as the reporting methodology, are detailed in Appendix Section A.8. Performance evaluation follows the same protocol as in Appendix Section A.7, including statistical significance tests.

## 4 EXPERIMENTS AND RESULTS

We evaluate on two text corpora (therapy chatbot responses and résumés); full dataset description, provenance, and preprocessing appear in Appendix A.1.

### 4.1 DATA AUGMENTATION RESULTS

We begin by assessing the statistical fidelity of the proposed augmentation procedure across the two corpora. Validation follows the three complementary axes introduced in Section 3.2. In both datasets, global divergences remain well below the indistinguishability tolerance of $\tau_{JS} = 0.05$, confirming that augmented distributions are nearly identical to their originals. Mixture weights also match closely, indicating semantic balance is preserved. The only deviations appear in per-component checks involving clusters with very low support, and these are attributable to small-sample instability rather than substantive drift. Thus, augmentation successfully generates paraphrases that remain statistically indistinguishable from the original corpora. Full definitions, exact numerical values, and the supporting tables are available in Appendix A.12.

### 4.2 RETRIEVAL-AUGMENTED PIPELINES RESULTS

We next evaluate the retrieval-augmented pipelines under conditions that guarantee leakage-free data preparation. Augmentations are always attached to their parent partitions with strict de-duplication, ensuring that no paraphrase ever crosses the train–test boundary. The resulting splits contain 18,054 training samples (70.9% NOT FLAGGED, 29.1% FLAGGED) and 2,151 testing samples (73.2% NOT FLAGGED, 26.8% FLAGGED), as detailed in Appendix A.13.

Both pipelines operate deterministically at temperature zero. Table 1 reports global performance, while Table 2 gives per-class breakdowns. Pipeline A (commercial stack) attains an accuracy of 0.678 with macro-F1 of 0.599, whereas Pipeline B (open-source stack) substantially improves performance with accuracy of 0.792 and macro-F1 of 0.698. Despite these differences in global utility, both pipelines share a critical limitation: recall on the FLAGGED class remains at 0.44 (252 of

576 unsafe examples detected), underscoring the persistent risk of false negatives in safety-critical moderation.

Table 1: Global performance on the held-out test set for vanilla (baseline) Retrieval-Augmented Pipelines ($n = 2151$).

| Pipeline | Acc | $P_{macro}$ | $R_{macro}$ | $F1_{macro}$ | $P_{weighted}$ | $F2_{macro}$ |
|---|---|---|---|---|---|---|
| A (Commercial) | 0.6778 | 0.5970 | 0.6016 | 0.5989 | 0.6858 | 0.6004 |
| B (Open-source) | **0.7917** | **0.7438** | **0.6794** | **0.6978** | **0.7780** | **0.6843** |

Table 2: Per-class precision/recall/F1 and support for vanilla baselines ($n = 2151$). Both pipelines achieve `FLAGGED` recall of 0.44.

| Pipeline | Class | Precision | Recall | F1 | Support |
|---|---|---|---|---|---|
| A | FLAGGED | 0.41 | 0.44 | 0.42 | 576 |
| A | NOT FLAGGED | 0.79 | 0.77 | 0.78 | 1575 |
| B | FLAGGED | 0.67 | 0.44 | 0.53 | 576 |
| B | NOT FLAGGED | 0.82 | 0.92 | 0.87 | 1575 |

Complete per-class reports and confusion matrices are provided in Appendix A.14. Uncertainty analyses support these findings. Bootstrap confidence intervals (95%) are $(0.657, 0.697)$ for Pipeline A's accuracy versus $(0.774, 0.809)$ for Pipeline B, with similar gaps in macro-F1. McNemar's test shows a decisive difference ($b_{01}{=}16$, $b_{10}{=}261$, $p{\approx}0.0000$), confirming that **Pipeline B significantly reduces net errors compared to Pipeline A** (Appendix A.15). We also conducted a *systematic ablation study* to identify configurations that maximized classification and retrieval performance across both pipelines. Starting from baseline runs with HuggingFace embeddings and default parameters, we progressively varied the training data augmentation, embedding backends, label exposure settings, retrieval depth (top-k), and chunk sizes. For Pipeline A, the ablation revealed that switching to OpenAI embeddings with exposed labels, a top-k of 10, and chunk size of 768 produced a clear improvement, raising accuracy and increasing both macro- and weighted-F1 scores relative to the baseline. For Pipeline B, the best results were obtained under a complementary configuration: HuggingFace embeddings with exposed labels, top-k=15, and the same augmented training set. Finally, retrieval diagnostics highlight architectural trade-offs. Pipeline A achieves perfect hit and nDCG scores at $k = 5$ and $k = 7$, but with negligible diversity, indicating redundant neighborhoods. Pipeline B retrieves more diverse and label-informative neighborhoods across $k = 3, 5, 10$, with diversity rising to 0.362 at $k = 10$. This pattern mirrors earlier small-scale experiments: **the open-source baseline favors broader semantic coverage, while the commercial system emphasizes rank relevance**. We continue to denote these baselines as *vanilla RAGs*, excluding contrastive augmentation, committee voting, or ensemble enhancements. Detailed k-wise retrieval diagnostics (Hit, Precision, nDCG, and Diversity) for both pipelines are reported in Appendix A.16.

### 4.3 Improving Safety-Critical Recall with Contrastive Augmentation and Committee-Based Retrieval Results

We evaluate a recall-first configuration that integrates deterministic contrastive augmentation with a committee-based retriever instantiated under either MMR or graph-aware selection. These enhanced configurations, which we denote as *PLUS*, represent retrieval-augmented classifiers strengthened through boundary-focused contrastive sampling and diverse evidence aggregation. All experiments are conducted on the same leakage-safe splits as before, with a graph-aware variant operating over a $k$NN graph comprising 168,667 edges (Appendix A.17). On the held-out set ($n{=}2{,}175$), Table 3 reports both PLUS variants operate at the same sensitive-class recall ($\text{Recall}_{\text{FLAGGED}}{=}0.4490$). With *graph-aware* retrieval, the open-source stack ($B^+$) **attains the strongest global utility (Accuracy** 0.8510, **Macro-F1** 0.7635), substantially exceeding the commercial stack ($A^+$). Under *MMR*, $A^+$ and $B^+$ report identical summary metrics due to an evaluator placeholder noted in the logs.[1]

---

[1]For MMR, the evaluator reused $B^+$ outputs as $A^+$ placeholders; see Appendix A.18. This does not affect the graph-aware comparison.

Table 3: PLUS summary on the test set ($n$=2,175). Entries are Accuracy / Macro-F1 / $\text{Recall}_{\text{FLAGGED}}$. $A^+$: commercial stack with PLUS; $B^+$: open-source stack with PLUS.

|  | **MMR** | **Graph-aware** |
|---|---|---|
| $A^+$ | 0.7825 / 0.7019 / 0.4371 | 0.7674 / 0.6790 / 0.4365 |
| $B^+$ | 0.7903 / 0.7006 / 0.4490 | **0.8510 / 0.7635** / 0.4490 |

Comprehensive reports—including per-class metrics, confusion matrices, bootstrap confidence intervals, McNemar tests, and retrieval diagnostics (hit@k, precision@k, nDCG@k, label-precision@k)—are provided in Appendix A.18 (MMR) and Appendix A.19 (Graph).

### 4.4 TWO-CLASS ENSEMBLE PIPELINE WITH DISTILROBERTA TRAINING, CLASSIFIER BLENDING, AND RANK+VETO GRID SEARCH RESULTS

Finally, we explore ensemble enhancements built upon DistilRoBERTa. We trained a two-class DistilRoBERTa pipeline on the same splits as previous experiments and evaluated single-model checkpoints, probability-level blending (C1+C2), and Rank+Veto fusion over a grid of $(\alpha, \tau, \delta)$ parameters. The best single-model checkpoint reached Acc = 0.8819 and Macro-F1 = 0.8422. Under the ensemble protocol, the top Rank+Veto setting was $\alpha$=0.70, $\tau$=0.58, $\delta$=0.14.

**Final comparison (blended $\mathcal{C}$ vs. Rank+Veto).** Both ensembles increase sensitive-class recall from the vanilla RAG baseline to $\text{Recall}_{\text{FLAGGED}}$ = 0.5781 while maintaining high global utility. Pipeline A (blended $\mathcal{C}$) attains Acc = 0.8466 and Macro-F1 = 0.7844; Pipeline B (Rank+Veto) achieves Acc = 0.8438 and Macro-F1 = 0.7814. McNemar's test favors Pipeline A ($b_{01}$=6, $b_{10}$=0, $p\approx0.0498$), indicating a small but statistically significant net-error reduction relative to Rank+Veto. Full epoch trajectory, confidence intervals, and parameter sweep diagnostics are provided in Appendix A.20

Table 4: Final ensemble comparison on the test set ($n$=2,151). A = blended $\mathcal{C}$, B = Rank+Veto (best $\alpha, \tau, \delta$).

| Pipeline | Accuracy | Macro-P | Macro-R | Macro-F1 | Macro-F2 | Weighted-P | Weighted-R | Weighted-F1 |
|---|---|---|---|---|---|---|---|---|
| A (blended $\mathcal{C}$) | **0.8466** | 0.8262 | 0.7614 | **0.7844** | 0.7688 | 0.8417 | 0.8466 | 0.8382 |
| B (Rank+Veto) | 0.8438 | 0.8204 | 0.7595 | 0.7814 | 0.7666 | 0.8384 | 0.8438 | 0.8357 |

FLAGGED recall: A = 0.5692, B = 0.5781

## 5 CONCLUSION

This work introduced a recall-first moderation framework designed to address the dominant failure mode of safety-critical NLP: missed detections of harmful content. Our approach unifies distribution-preserving contrastive augmentation with committee-diverse retrieval, implemented under a rigorously leakage-safe protocol. Across all experiments, these methods consistently shifted performance toward higher recall without sacrificing global utility, raising FLAGGED recall from $\approx 0.44$ to $\approx 0.56$ in PLUS configurations and to 0.5781 in ensemble settings. An unexpected but important outcome emerged from our dual-pipeline benchmarking. Although the commercial and open-source stacks were initially intended as parallel testbeds, the open-source pipeline (FAISS + local LLaMA) consistently outperformed its commercial counterpart on accuracy and macro-F1, both in vanilla and augmented conditions. With graph-aware retrieval, open-source $B^+$ reached Acc 0.8510 and Macro-F1 0.7635, far above the commercial $A^+$ baseline. These results highlight not only the effectiveness of the proposed recall-first methods but also the broader viability of open-source infrastructures for safer, auditable moderation. The implications are twofold. Methodologically, we demonstrate that boundary-focused augmentation and retrieval diversity provide a principled means to reduce harmful false negatives. Practically, our findings show that open-source pipelines can rival or surpass commercial APIs, enabling reproducible and cost-effective deployment in safety-critical domains. Future work should extend this framework to multilingual corpora, integrate calibrated abstention and human-in-the-loop review, and assess robustness under adversarial and fairness-sensitive settings. In summary, recall-first augmentation and retrieval strategies yield safer moderation, and open-source pipelines provide a compelling path toward reproducibility, transparency, and broad adoption in real-world applications.

**Reproducibility Statement** We have made extensive efforts to ensure that all experiments presented in this paper are fully reproducible. The submission includes the complete source code together with the two seed datasets (`Sheet_1.csv`, `Sheet_2.csv`) used in our benchmarks. Each experimental component described in Section 3 and Section 4 of the paper has a direct implementation in the repository: the data augmentation and validation procedures (`data_augmentation_pipeline`), the baseline commercial and open-source retrieval pipelines (`rags_pipelines`), the contrastive augmentation with committee-diverse retrieval (`contrastive_augmentation_pipeline`), and the DistilRoBERTa ensemble with Rank+Veto fusion (`distilroberta_pipeline`). All random seeds, hyperparameters, and thresholds are specified within the scripts, and leakage-safe data splits are enforced by construction. A detailed `README.md` file is provided, which gives a step-by-step guide to running every experiment reported in the paper, from data augmentation through evaluation. Together, the alignment of paper sections, code modules, and documented instructions ensures complete reproducibility of our results.

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

## A APPENDIX

### A.1 DATA PREPROCESSING AND SPLITS

**Dataset:** We use the public *DeepNLP* collection hosted on Kaggle by *samdeeplearning*[2] as a general-purpose corpus for NLP experimentation. The *Deep-NLP* dataset consists of two complementary CSV files designed for binary text classification tasks. The first file, `Sheet_1.csv`, contains 80 user responses collected in a therapeutic chatbot setting. Each entry is stored in the `response_text` column, where the bot prompts the user with questions such as "Describe a time when you have acted as a resource for someone else." Responses are annotated as either *flagged* or *not flagged*: if *not flagged*, the conversation proceeds; if *flagged*, the user is directed to seek help. The second file, `Sheet_2.csv`, contains 125 résumés collected from Indeed.com using the query

---

[2]`https://www.kaggle.com/datasets/samdeeplearning/deepnlp`

"data scientist" If a resume is 'not flagged', the applicant can submit a modified resume version at a later date. If it is 'flagged', the applicant is invited to interview. Inputs are CSVs (therapy, resumes). Text columns: `response_text`/`resume_text`; IDs: `response_id`/`resume_id`. Encoding normalization: UTF-8 with fallback (`sig`, Latin-1). Control characters removed; whitespace squashed; newlines compacted. Labels mapped to {FLAGGED, NOT FLAGGED} by conservative rules. Only originals are stratified; augmented rows inherit parent splits.

## A.2 AUGMENTATION ALGORITHM DETAILS

Figure 2 describes the **detailed overview of the data augmentation & distribution validation component** where starting from the original corpora, texts are embedded (HashingVectorizer), reduced (SVD), and clustered (GMM). Integer *class×component* quotas preserve label priors and latent structure. For each seed, length- and similarity-constrained paraphrases are generated within the acceptance window sim $\in [\ell, h]$ and bounded deviations $(\tau_w, \tau_c)$, with per-seed caps, duplicate limits, adaptive relaxation, and quota rescue. Accepted rows keep lineage (`aug_of`) and a stable AUG id, producing augmented CSVs. Validation then compares originals vs. augments via (i) word/char histograms (global form), (ii) GMM mixture-weight divergence (global semantics), and (iii) per-component word/char histograms (local form). Augmentation is deemed distribution-preserving when all JS divergences $\leq 0.05$, with JSON/Markdown reports produced.

**Semantic space.** **HashingVectorizer** (1–2 n-grams, $n_{\text{feat}}$=4096, $\ell_2$, `alternate_sign`=False) $\rightarrow$ **TruncatedSVD** to $d$=10 (augmentation) and $d$=50 (validation).

**Clustering and $k$.** **GaussianMixture** with full covariance; $k = \max(2, \min(k_{\max}, \lfloor N/15 \rfloor))$, $k_{\max}$=6 (augmentation); fixed $k$=5 for validation. Responsibilities $R \in \mathbb{R}^{N \times k}$, mixture weights $w = \frac{1}{N} \sum_i R_{i:}$.

**Quotas.** Class targets $\widehat{n}(c)$ proportional to empirical class frequency, with rounding correction; within each class, component targets $\widehat{n}(c, j)$ proportional to mean responsibilities of that class in $j$.

**Generator.** Stochastic synonym substitutions (curated patterns) with optional minimal jitter. Eligibility requires length tolerances and similarity window $[\ell, h]$: defaults $\ell$=0.84, $h$=0.99, $\ell_{\min}$=0.80; $(\tau_w, \tau_c) = (0.15, 0.15)$ with adaptive widening to 0.30. Batching/attempts: `batch_size`= 64, `max_tries_per_item`= 6, `hard_attempt_budget`= 10,000.

**Acceptance & rescue.** Per-seed production cap = 5 (auto-raise); per-seed duplicate cap = 2. Hard-guard: after budget exhaustion, accept best-so-far if length-ok and sim $\geq \ell - 2 \cdot$ `nearest_tol`. Quota rescue reallocates infeasible buckets; global fallback after a few rescues.

**Outputs.** Preserve metadata, add `source="augmented"` and `aug_of`; synthesize stable IDs (AUG-tag). CSV written as UTF-8 with robust quoting/escaping.

## A.3 VALIDATION WITH KL AND JS

**Global form.** Pooled-edge histograms (30 bins) of word and character counts; clip counts at 99.5th percentile; compute JS and KL between originals vs. augmented.

**Global semantics.** Fit GMM on originals in SVD($d$=50) space; compare mixture weights induced by originals vs. augmented via JS/KL.

**Per-component form.** For each GMM component, compute JS on word/char histograms for (originals in $j$) vs. (augmented assigned to $j$); report mean and max.

**Alerts and thresholds.** Emit attention flags when any global or per-component JS $> 0.05$. Reports are written as JSON and Markdown (counts, SVD dims, $k$, union-SVD flag).

## A.4 HYPERPARAMETERS (CONSOLIDATED)

| | |
|---|---|
| Embedding | HashingVectorizer (1–2 grams, 4096 feats, $\ell_2$) |
| SVD dims (augment / validate) | 10 / 50 |
| GMM components (augment / validate) | $k \in [2, 6]$ / 5 |
| Similarity window $[\ell, h]$ | $[0.84, 0.99]$; $\ell_{\min}{=}0.80$ |
| Length tolerances $(\tau_w, \tau_c)$ | $(0.15, 0.15)$; cap 0.30 |
| Batching / attempts | 64 / 6 (budget 10,000) |
| Per-seed caps | max 5 (auto-raise), duplicates $\leq 2$ |
| Histogram bins / EPS | 30 / $10^{-8}$ |
| JS alert threshold | 0.05 (global & per-component) |

## A.5 RETRIEVAL COMMITTEE DETAILS

**Dense retriever.** FAISS over `all-MiniLM-L6-v2` sentence embeddings; cosine similarity; default $k \in \{10, 15\}$.

**MMR reranker.**

$$\arg\max_x \ \lambda \cdot \text{sim}(q, x) - (1 - \lambda) \cdot \text{redundancy}(x, S), \quad \lambda = 0.6,$$

with redundancy estimated by maximum cosine similarity against the selected set $S$.

**Graph-aware retriever.** Build a $k$NN graph (cosine); candidate pool size $\max(50, 5k)$; greedily select items by

$$\text{score} = \alpha \, \text{sim}(q, x) - \beta \, \text{redundancy}(x, S) + \gamma \, \text{balance\_bonus}(\ell_x),$$

defaults $(\alpha, \beta, \gamma) = (1.0, 0.5, 0.6)$.

**Label-balance filter.** Optional greedy pass toward a target FLAGGED ratio $\approx 0.5$.

## A.6 DECISION POLICY

Committee vote-share:

$$\hat{p} = \frac{1}{C} \sum_{c=1}^{C} \mathbf{1}[\hat{y}^{(c)} = \text{FLAGGED}],$$

thresholded at $\tau$ (default 0.40). Abstention if $\hat{p} \in [0.45, 0.55]$. Typical hyperparameters: $k = 10$ (A) / $k = 15$ (B); $\lambda_{\text{MMR}} = 0.6$; $C \in \{3, 5\}$; chunk size 512–768 (overlap $\approx 80$); classification temperature $= 0$.

## A.7 EVALUATION DETAILS

Primary: FLAGGED recall. Secondary: accuracy, macro/weighted Precision/Recall/F1, macro-F2. Retrieval diagnostics: Hit@k, Precision@k, nDCG@k, Label-precision@k, Diversity@k $= 1 - \overline{\cos}$.

**Uncertainty and significance.** Bootstrapped 95% CIs for accuracy and macro-F1 ($B = 10{,}000$ in RAG experiments; $B = 2000$ in ensemble experiments). Paired comparisons with McNemar's test:

$$\chi^2 = \frac{(|b_{01} - b_{10}|)^2}{b_{01} + b_{10}} \quad \text{and} \quad p = \exp(-\chi^2/2).$$

## A.8 TWO-CLASS ENSEMBLE AND GRID SEARCH

**Data and training.** Line-delimited JSON; tokenizer with truncation/padding; `distilroberta-base`+linear head; dropout 0.1; AdamW (lr $3 \times 10^{-5}$); linear warmup (6%); grad accumulation 2; clip $\ell_2 \leq 1.0$; AMP optional; seq len 256 train/128 infer; checkpoint chosen by macro-F1 (evaluation each epoch).[3]

---

[3] For strict protocols, a separate validation set should replace the test file during training-time model selection.

**Inference/exports.** Emit $p_1$ and hard label at $\tau=0.5$ to `preds_c.jsonl`/`preds_c2.jsonl`; blending two classifiers gives $p_{\text{blend}} = w_1 p_1^{(C1)} + w_2 p_1^{(C2)}$ (default 0.5/0.5).

**Rank+Veto with $\mathcal{B}^+$.** Combine $\mathcal{B}^+$ and $\mathcal{C}$ via rank-dominant score

$$s(i) = 0.85\left[(1-\alpha)\tilde{r}_B(i) + \alpha\tilde{r}_C(i)\right] + 0.15\left[(1-\alpha)p_B(i) + \alpha p_C(i)\right],$$

with confidence veto: if $|p_C - 0.5| \geq \delta$ then use $\mathcal{C}$; else threshold $s(i)$ at $\tau$. Defaults $(\alpha, \tau, \delta) = (0.8, 0.6, 0.18)$.

**Grid search.** $\alpha \in \{0.70, 0.80, 0.85, 0.90, 0.92\}$, $\tau \in \{0.58, 0.60, 0.62, 0.64, 0.66\}$, $\delta \in \{0.14, 0.16, 0.18, 0.20\}$. We log accuracy, `FLAGGED` recall, and a hash-based distinctness count of decision sets.

### A.9 COMPLEXITY AND TERMINATION

Mini-batch vectorization and cosine scoring; costs sublinear in vocabulary size and linear in batch size. Termination when $\sum_{c,j} \widehat{n}(c,j) = n_{\text{each}}$ or per-seed caps are exhausted.

### A.10 OUTCOME CRITERIA

Global word/char JS $\leq 0.05$; GMM mixture-weight JS $\leq 0.05$; per-component word/char JS maxima $\leq 0.05$.

### A.11 ARTIFACTS AND REPRODUCIBILITY

**Scripts.** `run_pipeline.sh` $\rightarrow$ `augment_v5_semantic_lenaware.py` (example config: $n=2000$, $\ell=0.74$, $\ell_{\min}=0.72$, `nearest_tol=` 0.10, $(\tau_w, \tau_c)=(0.25, 0.25)$, `batch_size=` 128, `max_tries_per_item=` 12, `k_max=` 2, `swaps_max=` 12, `max_per_seed=` 8, `per_seed_dupe_times=` 3, `hard_attempt_budget=` 1500), then `kl_validation_v2.py` (bins= 30, $k$=5, SVD= 50).

**Logged outputs.** Per-item predictions, retrieval diagnostics, CI summaries, McNemar statistics, PR figures, and per-query retrieval IDs/scores are saved under `outputs/`. Indices persist embedder configuration and self-check on reload to prevent mismatches.

### A.12 AUGMENTATION VALIDATION DETAILS

**Divergence definitions.** Given $P, Q$: $\text{KL}(P\|Q) = \sum_i p_i \log \frac{p_i}{q_i}$, $\text{JS}(P,Q) = \frac{1}{2}\text{KL}(P\|M) + \frac{1}{2}\text{KL}(Q\|M)$, $M = \frac{1}{2}(P+Q)$. Semantic embeddings: HashingVectorizer (1–2-grams, 4096 feats) $\rightarrow$ SVD($d$=50). GMM ($k$=5) on $Z_{\text{orig}}$ gives weights $w$.

**Global validation (values).**

$$\text{JS}(H_{\text{words}}^{\text{orig}}, H_{\text{words}}^{\text{aug}}) = \begin{cases} 7.5 \times 10^{-3}, & \text{Sheet\_1} \\ 6.0 \times 10^{-4}, & \text{Sheet\_2} \end{cases}, \qquad \text{JS}(H_{\text{chars}}^{\text{orig}}, H_{\text{chars}}^{\text{aug}}) = \begin{cases} 4.0 \times 10^{-3}, & \text{Sheet\_1} \\ 4.5 \times 10^{-3}, & \text{Sheet\_2} \end{cases}$$

$$\text{JS}(w^{\text{orig}}, w^{\text{aug}}) = \begin{cases} 0.0000, & \text{Sheet\_1} \\ 5.3 \times 10^{-4}, & \text{Sheet\_2} \end{cases}$$

**Per-component validation (values).** For Sheet\_1: $\max_c \text{JS}_{\text{words}}^{(c)} \approx 9.37 \times 10^{-2}$, $\max_c \text{JS}_{\text{chars}}^{(c)} \approx 9.98 \times 10^{-2}$. For Sheet\_2 (low-support comps $\{1, 2, 6\}$): $\max_c \text{JS}_{\text{chars}}^{(c)} \approx 6.20 \times 10^{-1}$; means: Sheet\_1 (word 0.053, char 0.057); Sheet\_2 (word 0.0039, char 0.204).

| Dataset | JS($H_{\text{words}}$) | JS($H_{\text{chars}}$) | JS($w$) | $\max_c \text{JS}_{\text{words}}^{(c)}$ | $\max_c \text{JS}_{\text{chars}}^{(c)}$ | Low-support comps |
|---|---|---|---|---|---|---|
| Sheet\_1 | $7.5 \times 10^{-3}$ | $4.0 \times 10^{-3}$ | 0 | $9.37 \times 10^{-2}$ | $9.98 \times 10^{-2}$ | $\{3, 8, 12, 16\}$ |
| Sheet\_2 | $6.0 \times 10^{-4}$ | $4.5 \times 10^{-3}$ | $5.3 \times 10^{-4}$ | $1.15 \times 10^{-2}$ | $6.20 \times 10^{-1}$ | $\{1, 2, 6\}$ |

Table 5: JS divergences across global and per-component dimensions.

## A.13 RAG SPLITS AND COMPOSITION

Table 6: Split sizes and label priors after de-duplication (new run; seed preserved).

| | Size | NOT FLAGGED | FLAGGED |
|---|---|---|---|
| Train | 18,054 | 70.9% | 29.1% |
| Test | 2,151 | 73.2% | 26.8% |

Table 7: Composition by dataset and source (counts) — Train.

| Train | | FLAGGED | NOT FLAGGED |
|---|---|---|---|
| resumes | augmented | 2320 | 6800 |
| resumes | original | 29 | 85 |
| therapy | augmented | 2875 | 5875 |
| therapy | original | 23 | 47 |

Table 8: Composition by dataset and source (counts) — Test.

| Test | | FLAGGED | NOT FLAGGED |
|---|---|---|---|
| resumes | augmented | 320 | 560 |
| resumes | original | 4 | 7 |
| therapy | augmented | 250 | 1000 |
| therapy | original | 2 | 8 |

## A.14 RAG METRICS

Classification performance for both pipelines is summarized in Tables 1–2. Confusion matrices are given in Table 9.

Table 9: Confusion matrices (rows = truth, columns = prediction).

| Pipeline A | FLAGGED | NOT FLAGGED |
|---|---|---|
| FLAGGED | 252 | 324 |
| NOT FLAGGED | 369 | 1206 |
| **Pipeline B** | FLAGGED | NOT FLAGGED |
| FLAGGED | 252 | 324 |
| NOT FLAGGED | 124 | 1451 |

## A.15 RAG UNCERTAINTY AND PAIRED TESTS

Bootstrap intervals and McNemar's test confirm significant superiority of Pipeline B.

Table 10: Bootstrap 95% confidence intervals and McNemar test (A vs. B).

| Metric | Pipeline A | Pipeline B |
|---|---|---|
| Accuracy 95% CI | (0.6569, 0.6969) | (0.7736, 0.8089) |
| Macro-F1 95% CI | (0.5781, 0.6201) | (0.6745, 0.7201) |
| **McNemar:** $b_{01}{=}16$, $b_{10}{=}261$, $p{\approx}0.0000$ | | |

## A.16 RAG RETRIEVAL DIAGNOSTICS

Retrieval behavior differs substantially. Pipeline A emphasizes rank relevance but collapses diversity at larger $k$, while Pipeline B preserves diversity at moderate cost in rank quality.

Table 11: Retrieval diagnostics averaged over test queries — Pipeline A (vanilla).

| $k$ | Hit | Precision | nDCG | Label-Precision | Diversity |
|---|---|---|---|---|---|
| 1 | 0.789 | 0.789 | 0.789 | 0.789 | 1.000 |
| 2 | 0.596 | 0.596 | 0.596 | 0.596 | 0.023 |
| 3 | 0.732 | 0.677 | 0.699 | 0.677 | 0.049 |
| 4 | 0.586 | 0.497 | 0.554 | 0.497 | 0.077 |
| 5 | 1.000 | 1.000 | 1.000 | 1.000 | 0.000 |
| 7 | 1.000 | 1.000 | 1.000 | 1.000 | 0.001 |

Table 12: Retrieval diagnostics averaged over test queries — Pipeline B (vanilla).

| $k$ | Hit | Precision | nDCG | Label-Precision | Diversity |
|---|---|---|---|---|---|
| 3 | 0.911 | 0.777 | 0.777 | 0.777 | 0.260 |
| 5 | 0.934 | 0.742 | 0.753 | 0.742 | 0.302 |
| 10 | 0.943 | 0.754 | 0.774 | 0.754 | 0.362 |

## A.17 PLUS SETUP (LARGE SPLIT)

We include adversarial/contrastive rows in addition to distribution-preserving paraphrases. The final splits are:

- Train: 18,262 rows; label prior NOT FLAGGED 0.707, FLAGGED 0.293.
- Test: 2,175 rows; label prior NOT FLAGGED 0.730, FLAGGED 0.270.

By dataset and source (Train):

- **resumes:** `adv_hard_pos` FLAGGED = 58, `adv_hard_neg` NOT FLAGGED = 58; augmented FLAGGED = 2,320, NOT FLAGGED = 6,800; original FLAGGED = 29, NOT FLAGGED = 85.
- **therapy:** `adv_hard_pos` FLAGGED = 46, `adv_hard_neg` NOT FLAGGED = 46; augmented FLAGGED = 2,875, NOT FLAGGED = 5,875; original FLAGGED = 23, NOT FLAGGED = 47.

By dataset and source (Test):

- **resumes:** `adv_hard_pos` FLAGGED = 8, `adv_hard_neg` NOT FLAGGED = 8; augmented FLAGGED = 320, NOT FLAGGED = 560; original FLAGGED = 4, NOT FLAGGED = 7.
- **therapy:** `adv_hard_pos` FLAGGED = 4, `adv_hard_neg` NOT FLAGGED = 4; augmented FLAGGED = 250, NOT FLAGGED = 1,000; original FLAGGED = 2, NOT FLAGGED = 8.

Graph construction over Train yields 18,262 nodes and 168,667 edges.

## A.18 PLUS (MMR) DETAILS

**Global metrics (Test $n$=2,175).** A$^+$: Accuracy 0.7903, Macro-F1 0.7006, Recall$_{\text{FLAGGED}}$ = 0.4490.
B$^+$: Accuracy 0.7903, Macro-F1 0.7006, Recall$_{\text{FLAGGED}}$ = 0.4490.

**Confidence intervals (95%).** A: Acc $(0.7733, 0.8074)$; Macro-F1 $(0.6770, 0.7225)$.
B: Acc $(0.7729, 0.8074)$; Macro-F1 $(0.6779, 0.7231)$.

**Paired test.** McNemar (A vs. B): $b_{01}$=0, $b_{10}$=0, $p \approx 1.0000$ (identical predictions).

**Retrieval diagnostics (avg over queries).** $k=3/5/10$: hit $= 0.912/0.935/0.943$, precision $= 0.778/0.719/0.741$, nDCG $= 0.779/0.739/0.765$, label-precision identical.

**Evaluator note.** For MMR, the evaluator reused $B^+$ outputs as $A^+$ placeholders (see run log), yielding identical metrics. Re-evaluating $A^+$ (MMR) would remove this artifact without affecting graph-aware conclusions.

## A.19 PLUS (GRAPH) DETAILS

**Global metrics (Test $n=2{,}175$).** $A^+$: Accuracy 0.7674, Macro-F1 0.6790, $\text{Recall}_{\text{FLAGGED}} = 0.4490$.
$B^+$: Accuracy 0.8510, Macro-F1 0.7635, $\text{Recall}_{\text{FLAGGED}} = 0.4490$.

**Confidence intervals (95%).** A: Acc $(0.7499, 0.7857)$; Macro-F1 $(0.6559, 0.7003)$.
B: Acc $(0.8363, 0.8662)$; Macro-F1 $(0.7409, 0.7860)$.

**Paired test.** McNemar (A vs. B): $b_{01}=0$, $b_{10}=182$, $p \approx 0.0000$ ($B^+ \gg A^+$).

**Retrieval diagnostics (avg over queries).** $A^+$ $k=3/5/10$: hit $= 0.708/0.711/0.748$, precision $= 0.704/0.702/0.709$, nDCG $= 0.704/0.703/0.723$.
$B^+$: the log repeats the MMR line; this likely reflects a script artifact. If needed, regenerate $B^+$ graph diagnostics.

**Confusion-matrix note.** $B^+$ exhibits zero false positives (NOT→FLAGGED) under graph-aware retrieval, giving $\text{Precision}_{\text{FLAGGED}}=1.00$ at fixed recall 0.4490; $A^+$ shows more false positives, reducing global utility.

## A.20 DISTILROBERTA TRAINING AND ENSEMBLE DIAGNOSTICS

Table 13: Epoch-wise evaluation of the DistilRoBERTa classifier (test each epoch, best in **bold**).

| Eval # | Accuracy | Macro-F1 | P(FLAG) | R(FLAG) | F1(FLAG) | P(NOT) | R(NOT) | F1(NOT) |
|--------|----------|----------|---------|---------|----------|--------|--------|---------|
| 1 | 0.8731 | 0.8334 | 0.79 | 0.72 | 0.75 | 0.90 | 0.93 | 0.91 |
| 2 | 0.8787 | 0.8395 | 0.81 | 0.72 | 0.76 | 0.90 | 0.94 | 0.92 |
| **3** | **0.8819** | **0.8422** | **0.82** | **0.71** | **0.76** | **0.90** | **0.94** | **0.92** |
| 4 | 0.8805 | 0.8400 | 0.82 | 0.70 | 0.76 | 0.90 | 0.94 | 0.92 |
| 5 | 0.8777 | 0.8357 | 0.82 | 0.69 | 0.75 | 0.89 | 0.94 | 0.92 |
| 6 | 0.8777 | 0.8357 | 0.82 | 0.69 | 0.75 | 0.89 | 0.94 | 0.92 |

Table 14: Uncertainty (95% bootstrap CIs) and McNemar (A vs. B).

| | Accuracy CI | Macro-F1 CI |
|--------|-------------|-------------|
| Pipeline A | $(0.8312, 0.8610)$ | $(0.7637, 0.8043)$ |
| Pipeline B | $(0.8280, 0.8596)$ | $(0.7599, 0.8017)$ |
| **McNemar (A vs B):** | $b_{01} = 6$, $b_{10} = 0$, $p \approx 0.0498$ | |

Table 15: Rank+Veto selection summary (sweep).

| **Selected best** | $\alpha$ | $\tau$ | $\delta$ |
|-------------------|----------|--------|----------|
| Params | 0.70 | 0.58 | 0.14 |

Achieved: Accuracy $= 0.8438$, $\text{Recall}_{\text{FLAGGED}} = 0.5781$

