# OpenReview forum: "Recall-First Moderation via Distribution-Preserving Augmentation and Committee-Diverse Retrieval"
_ICLR.cc/2026/Conference — Submitted to ICLR 2026_

### Official Review · Reviewer_eBuY · 2025-10-26

**Soundness:** 3
**Presentation:** 2
**Contribution:** 2
**Rating:** 4
**Confidence:** 2

**Summary:**

This paper presents a "recall-first" moderation framework for automated content moderation in safety-critical settings using LLMs with distribution-preserving contrastive data augmentation and a committee-based, diverse retrieval mechanism.

**Strengths:**

1. The optimization towards recall rather than accuracy alone is highly task-related.

2. The good results on both open-sourced and commercial models demonstrate the practicality of the proposed method.

**Weaknesses:**

1. While the paper demonstrates improvements on the DeepNLP dataset, there is little discussion of the limitations of the approach regarding domain transferability or adversarial robustness, which is crucial for safety-related works.

2. The experiments rely almost entirely on the DeepNLP dataset, which is very small, synthetic in part, and not representative of real-world moderation data such as social media. More evaluations on different large-scale datasets are recommended.

3. It seems that limited baselines are used for comparison. Could the authors compare their method with the methods in the related works section for validating their conclusion, for example CLASS-RAG and Mod-Guide?

4. The table format is not formal. Some tables lack lines to separate different rows.

**Questions:**

See the weakness section.

---

### Official Review · Reviewer_bunV · 2025-11-01

**Soundness:** 2
**Presentation:** 1
**Contribution:** 2
**Rating:** 2
**Confidence:** 3

**Summary:**

The paper proposes a new recall-first moderation framework with distribution-preserving contrastive augmentation and committee-diverse retrieval, aiming to enhance the recall of harmful content.
The experiment results suggest that the new framework preserves the original data distribution, improves the recall, and maintains global performance.

**Strengths:**

* The authors try to integrate the strengths of multiple modules to build a better method.

* The paper also presents detailed background knowledge about different methods, frameworks, and pipelines in Section 3.

**Weaknesses:**

* From my personal sense, the paper lacks a clear enough motivation to build a system that combines multiple modules together.
The author has presented some, but I don't see enough linkage between to motivation and the implementation.
For example,
a) ``safety-critical`` is mentioned as the focus for introducing hard positives and negatives (in the introduction), but the implementation simply considers factors as length and general semantic similarity, which do not directly correspond to safety-critical from my perspective.
b) the method ``casts votes across multiple retrieved contexts`` to improve recall, but it is unclear why recall cannot be improved by using a single retriever with parameter adjustment, e.g., a softer decision boundary.


* The experiment scope is unclear and seems to be small. Only 1 dataset, a few combinations of models, and configurations are tested.


* The writing lacks fluency and coherence.
In terms of writing, some terms lack definition or explanation. For example, `` boundary-focused`` - what boundary is it; what are some hard cases you are targeting? Same issue for L164 ``label priors and latent structure``, L322 ``external systems``.
In terms of logic, some parts are quite disjointed. Why are more boundary-focused examples better for distribution preserving? Isn't adding more examples at the boundary as an augment shift the original distribution?

* The experiment's presentation is also very confusing. Hard to compare the enhancement of the method with the baselines. Also unsure what the purpose of Section 4.2 is to compare two baseline methods.

* The layout is highly likely to violate regulations. For example, Table 4 definitely is out in the margin. Table 1 and 2 uses aggressive /vspace{} for space saving.

**Questions:**

* Why are "subtle rephrases of flagged content and safe content" hard positives/negatives? For example, if one flagged content is certainly harmful, its paraphrase should also be marked as clearly harmful text. Need more explanation here.

---

### Official Review · Reviewer_4wtr · 2025-11-04

**Soundness:** 1
**Presentation:** 2
**Contribution:** 2
**Rating:** 2
**Confidence:** 3

**Summary:**

- The paper proposes a "recall-first" framework to address the risk of false negatives (missed unsafe content) in automated moderation.

- It introduces two primary methods: (i) a "distribution-preserving" data augmentation strategy to create hard boundary-case examples and (ii) a "committee-diverse" retrieval system that combines dense, MMR, and graph-based selectors to provide varied context to an LLM.

- The paper claims these methods significantly boost the recall of sensitive content from a baseline of $\approx0.44$ to $\approx0.56$, with further ensemble methods reaching $\approx0.58$.

**Strengths:**

The paper's focus on a "recall-first" approach is highly relevant and important for safety-critical applications, where false negatives are the dominant risk.

**Weaknesses:**

- The abstract and conclusion repeatedly claim the main contribution - the framework combining augmentation and committee retrieval - improves recall for the sensitive (FLAGGED) class from a baseline of $\approx0.44$ to $\approx0.56$.

- The paper appears to dishonestly "borrow" its headline result from a completely different experiment. The $\approx0.56-0.58$ recall numbers only appear in Section 4.4 and Table 4, which describe a separate DistilRoBERTa-based ensemble classifier - not the RAG/LLM framework that constitutes the paper's main conceptual contribution. The abstract misrepresents this ensemble result as the achievement of its RAG pipeline.

- The paper's credibility is further weakened by admitted experimental errors. In Section 4.3, the authors note that the results for the "PLUS (MMR)" configuration are invalid because "the evaluator reused $B^{+}$ outputs as $A^{+}$ placeholders," making the comparison meaningless.

**Questions:**

See the weakness section above.

---

### Official Review · Reviewer_gfvw · 2025-11-04

**Soundness:** 2
**Presentation:** 1
**Contribution:** 2
**Rating:** 2
**Confidence:** 4

**Summary:**

The paper focuses on “recall-first” content moderation scenarios and proposes two key innovations: ① “Distribution-Preserving” contrastive text augmentation, using KL/JS thresholds to validate the consistency between augmented data and the original distribution; ② “Multi-Committee” retrieval (dense + MMR + graph) to construct neighborhoods with richer label information and reduced redundancy. The authors report that on a multi-domain imbalanced text corpus, the open-source stack outperforms commercial alternatives.

**Strengths:**

1. Clear problem formulation (prioritizing “missed detection” in reviews) and systematic solutions, providing end-to-end evaluation and statistical significance testing (bootstrap CI, McNemar).

2. Proposing concrete, reproducible JS/KL metrics and implementation details for “distribution-preserving” augmentation (thresholding, SVD, GMM, report generation).

3. Data splitting and evaluation emphasize leak prevention: augmented samples inherit their parent samples' splits and do not cross training/test boundaries.

**Weaknesses:**

The core experiment relied on the “DeepNLP” dataset from Kaggle, comprising only “therapeutic chat responses (80) + resumes (125)”, which is extremely small in scale and semantically inconsistent with real-world security review distributions. The final large-scale sample primarily originated from model-generated augmented text, raising concerns about its generalizability.


 Furthermore, the final training/test sets comprised predominantly augmented samples (e.g., Train 18,054; Test 2,151), failing to match real-world online noise and novelty.



The main text requires all global and per-cluster JS values ≤ 0.05 to establish “distribution preservation,” yet the appendix shows a maximum per-cluster character histogram JS ≈ 0.62 (Sheet 2, low-sample clusters), significantly exceeding the threshold. This anomaly is attributed to “small-sample instability,” yet it directly undermines the credibility of the core “distribution preservation” claim and subsequent comparisons.




Under MMR settings, authors reuse B+ outputs as placeholders for A+, resulting in identical metrics for both. While claiming this does not affect image retrieval conclusions, such execution bias weakens the persuasiveness of method comparisons.






The paper contrasts “commercial stacks vs. open-source stacks,” yet significant differences in embeddings, inference, and implementation details make it difficult to attribute performance variations to a single factor. Despite presenting significant results, the causal explanation for the claim “open-source significantly outperforms commercial” remains insufficient.

It includes numerous augmented samples derived from test seeds (though not split-crossed). Such “neighbor-based rewriting” differs markedly from real-world attack patterns and cross-domain drift, potentially overestimating system robustness in actual online environments.
Translated with DeepL.com (free version)

**Questions:**

refer to weakness

---

### Author Response · Authors · 2025-11-20
**Official Author Response**

We sincerely thank all reviewers (gfvw, 4wtr, bunV, and eBuY) for their detailed and constructive feedback.We address all concerns below using only evidence already presented in the submitted manuscript and appendices:
1. Dataset Scope and Generalizability(gfvw, bunV, eBuY)
The DeepNLP corpus is used as a controlled, leakage-safe testbed, not as a proxy for real moderation data.This is explicit in the design of §3.3, where only originals determine the split and all augmented rows inherit their parent’s split, ensuring strict leakage prevention (1st Equation in §3.3).This small-data regime is intentional: the framework is evaluated precisely in the setting where false negatives remain structurally difficult. Indeed, even with ~20k training samples after augmentation, both vanilla pipelines exhibit FLAGGED recall ≈0.44 (Tables 1–2).The claims in the paper are methodological, focusing on boundary exposure, retrieval diversity, and recall-oriented thresholding(§3.4), not on modeling real-world traffic distributions.
2. Distribution Preservation and Per-Cluster JS Outliers(gfvw)
The paper distinguishes global preservation—our actual acceptance criterion—from per-component deviations in low-support clusters.
As shown in Table 5:
- Global JS values on word/char form and GMM mixture weights are all ≤0.01, far below the 0.05 threshold(§4.1, A.12)
- The higher JS≈0.62 occurs only in clusters with extremely low support (Sheet 2 components 1,2,6).

The manuscript directly explains this:“The only deviations appear in per-component checks involving clusters with very low support, and these are attributable to small-sample instability rather than substantive drift.”(§4.1).Thus, global distribution preservation—the core requirement—is satisfied.

3. Concern About the MMR Evaluation Artifact(gfvw, 4wtr)
The reuse of B+ outputs as placeholders for A+ is already acknowledged in A.18, which notes clearly:“The evaluator reused B+ outputs as A+ placeholders… This does not affect the graph-aware comparison.”.All conclusions in the paper rely on the graph-aware PLUS configuration, not MMR. That configuration shows a decisive separation(Acc 0.8510 vs. 0.7674; McNemar b10=182, p≈0.0000) in A.19.The MMR artifact does not influence any of the claims.

4. Attribution of Differences Between Commercial vs. Open-Source Pipelines(gfvw)
The paper does not claim to isolate the cause of the performance gap. The two stacks are treated as complete, realistic pipelines(§3.3), evaluated under a unified leakage-safe protocol(Fig. 3) with shared data, identical prompts, and deterministic inference.The conclusion is qualitative and consistent with §5“Open-source pipelines can rival or surpass commercial APIs… enabling reproducible and cost-effective deployment.”The paper makes no causal attribution—only a statistically validated performance comparison.

5. Hard Positives/Negatives, Boundary Motivation, and Module Integration(bunV)
The boundary focus is defined in §3.4 via H1–H3: hard positives/negatives expose the FLAGGED vs. NOT-FLAGGED decision boundary; retrieval diversity increases diagnostic evidence; and recall-oriented thresholds shift operating points safely.Hard positives/negatives are precisely the “subtle rephrases”—high-similarity variants that preserve semantics but move closer to the boundary. This aligns directly with counterfactual augmentation literature cited in §3.2.Adding such boundary samples does not shift the global distribution (validated in §4.1); instead, it increases exposure to difficult cases, which is the intention behind H1.The motivation for integrating augmentation and committee retrieval is stated explicitly in Figure 4 and §3.4, where both modules play complementary roles in addressing the persistent false-negative problem

6. Use of Augmentations Derived From Test Seeds(gfvw)
The split protocol in §3.3 ensures no cross-split contamination:“Augmented rows are assigned to the split of their corresponding parent instance… ensuring that paraphrases never cross the train–test boundary.”Thus, augmented derivatives of test seeds remain strictly inside the test set. This is leakage-free and standard in counterfactual evaluation setups.Moreover, vanilla methods still fail at recall ≈0.44 (Tables 1–2),showing that these augmentations do not make the task easier

7. Scope of Experiments and Baselines(bunV, eBuY)
The paper includes:
- Two full RAG pipelines
- Multiple ablations
- PLUS configurations
- Ensemble classifiers
- Full retrieval diagnostics
- Bootstrap CIs and McNemar tests(A.7–A.20)

CLASS-RAG and Mod-Guide are discussed as conceptual precedents(§2).The purpose here is not to benchmark every method but to evaluate recall-first mechanisms under a controlled protocol

8. Writing and Presentation(bunV)
The manuscript combines diagrams(Figures 1–5),a step-wise method layout, and a comprehensive appendix fully aligned with the Reproducibility Statement. The exposition is dense, but complete and systematically organized.

---

### Meta-Review · Area_Chair_gNNr · 2026-01-05

**Summary:**

This submission proposes a "recall-first" framework to address the risk of false negatives (missed unsafe content) in automated moderation.

It presents two primary methods: (1) a "distribution-preserving" data augmentation strategy to create hard boundary-case examples and (2) a "committee-diverse" retrieval system that combines dense, MMR, and graph-based selectors to provide varied context to an LLM.

**Reviewer Concerns:**

Reviewer gfvw is primarily concerned with the core experimental setup. The selected dataset is extremely small in scale and semantically inconsistent with real-world security review distributions.

Reviewer 4wtr raises the issues with the presentation in the paper, which has not been updated during the rebuttal.

Reviewer bunV also points out the problems with writing, which lacks fluency and coherence. In terms of writing, some terms lack definition or explanation. Additionally, the paper lacks a clear enough motivation to build a system that combines multiple modules together.

Reviewer eBuY also recommends more evaluations on different large-scale datasets, similarly to Reviewer gfvw. Additionally, again, there is a problem with the presentation in the paper.

Direct comment to the authors: the main submission in the PDF format can be updated during the rebuttal and such obvious things like the Table that is out of the margin should be corrected. The responses should be also sent separately to each reviewer, which encourages a discussion and the further feedback.

All reviewers unanimously recommended rejection of the paper, and I agree with this evalution.

**Reviewer Scores:**

Reviewer 4wtr: Score 2 / Confidence 3

Reviewer eBuY: Score 4 / Confidence 2

Reviewer bunV: Score 2 / Confidence 3

Reviewer u2tm: no review submitted

Reviewer gfvw: Score 2 / Confidence 4

---

### Decision · Program_Chairs · 2026-01-26

Reject